# GiellaLT — a stable infrastructure for Nordic minority languages and beyond

**Flammie A Pirinen**  **Sjur N. Moshagen**  **Katri Hiovain-Asikainen**

Divvun, Department of Language and Culture

UiT Norgga árktalaš universitehta

Tromsø, Norway

flammie.pirinen@uit.no

sjur.n.moshagen@uit.no

katri.hiovain-asikainen@uit.no

## Abstract

Long term language technology infrastructures are critical for continued maintenance of language technology based software that is used to support the use of languages in the digital world. In the Nordic area we have languages ranging from well-resourced national majority languages like Norwegian, Swedish and Finnish as well as minoritised, unresourced and indigenous languages like the Sámi languages. We present an infrastructure that has been built in over 20 years time that supports building language technology and tools for most of the Nordic languages as well as many of the languages all over the world, with focus on Sámi and other indigenous, minoritised and unresourced languages. We show that one common infrastructure can be used to build tools from keyboards and spell-checkers to machine translators, grammar checkers and text-to-speech as well as automatic speech recognition.

## 1 Introduction

Language technology infrastructures are needed for long-term maintenance of linguistic data and NLP applications derived from it. Specifically in a Nordic context, we have a selection of languages with very different requirements, and all differ from those that are commonly assumed in other NLP contexts, e.g. English and handful of most resourced languages in the world. The languages in the Nordic area range from decently resourced Indo-European languages (Norwegian bokmål, Swedish, Danish and Icelandic) to moderately resourced Uralic languages (Finnish, Estonian) to all low and unresourced, minoritised languages (Sámi languages, all other Uralic languages, Faroese, Greenlandic). We have an infrastructure that supports all of these languages, with a

focus on the smaller and less resourced languages and specifically on the Sámi languages. The infrastructure we provide has been in use for over a decade and in this article we describe strategies and workflows that we have found successful. It currently supports over 100 languages, many outside of the Nordic region.

The technical infrastructure builds on the concept that we aim to separate the technological work: programming and engineering, from the linguistic work: lexicography, grammar building, corpus annotation etc. In this way, we enable linguists and native informants to work on the language data and the engineers build and maintain the technological solutions in a meaningful way where both the technological solutions and the linguistic data are kept up to date and functional. This workflow is important since both linguistic and technological sides present ongoing challenges to be kept up to date. Regarding the linguistic content, the language norms change and grow, new words and expressions enter the lexicon regularly and other words and expressions become outdated. In technology, operating systems and environments, programming languages and APIs change all the time, making the NLP tools built a few years ago not usable a few years later. The research question we solve with our infrastructure is, how both parts can be kept up to date while not burdening the people working with the parts with details irrelevant for their work.

In other words, the infrastructure contains linguistic data, and technological implementations to build end user NLP-based tools and software from it. The tools that we build nowadays include writing tools, such as spelling and grammar checkers and correctors, speech synthesis and recognition, machine translation, intelligent dictionaries and various linguistic analysis tools. The technological infrastructure is composed of tools like version control systems, build systems and automa-

tion of building and distribution of the NLP tools. The underlying technologies here have changed a lot in the past 20 years, and will undoubtedly keep evolving. In this article we take a look on some concepts that have both stayed stable or evolved to be part of the core tools for us. In the NLP scene, the world has changed a lot in past years as well, with the traditional knowledge-based methodology being gradually replaced by data-driven approaches; in the GiellaLT infrastructure we are still following the expert-driven knowledge-based approach as it continues to be the most appropriate for unresourced languages, but we do not cover this dichotomy in detail; for more details of this we refer to (Wiechetek et al., 2022) that discusses the issue extensively.

In the past 20 years we have built language resources for several Sámi languages starting from virtually nothing; Even though we had a number of non-digital resources available, these were far from exhaustive. This means that our work also included normative discussions, requests and suggestions to the language normative organs, error classifications, and grammatical descriptions of phenomena not included in grammar books. In several cases, these phenomena needed traditional linguistic research. Based on this experience we suggest workflows and usage patterns along the technical solutions of the infrastructure that are effective for long term maintenance of linguistic software in support of continued digital existence of human languages.

The contributions of this article are: We present a stable Nordic language technology infrastructure that has supported Nordic language technology development for 20 years, we describe the best current practices we have learned in the years and based on the current state of things we sketch the potential future developments.

## 2 Background

The infrastructure presented in this article has been developed and maintained for at least 20 years now. The infrastrucutre has been discussed previously in Nodalida some 10 years ago Moshagen et al. (2013). In this work we aim to show updates and prove that the system has well stood the test of time in supporting Nordic languages. On one hand everything has changed between the years; computers and mobile platforms, operating systems, programming environments, on the other

hand, many solutions have stayed usable: rule-based finite state morphologies, dictionaries and linguistic data.

The foundation for the work presented in this article is the multilingual infrastructure *GiellaLT*, which includes over 100 languages, including most nordic ones: the Sámi languages, Faroese, Finnish, Norwegian, Swedish, other Uralic languages and many more. Everything produced in the *GiellaLT* infrastructure is under free and open licences and freely available. The corpora are available with free licensing where possible. The infrastructure is split code-wise in three GitHub organisations: *GiellaLT* containing the language data for each language, *Divvun* containing language independent code for the infrastructure and various applications, and *Giellatekno* for corpus infrastructure. End user tools served by the Divvun group are at *divvun.no* & *divvun.org*, and tools served by the Giellatekno group at *giellatekno.uit.no*, both at *UiT Norway's Arctic University*.

We build systems that include lexical data as well as rules governing morphophonology, syntax and semantics as well as a number of application specific information, e.g. grammatical rules for grammar checking, phonetic rules for *Text-To-Speech* (TTS) and so forth.

The language-independent work is currently done within the infrastructure, the language-independent features and updates that are relevant to all languages are semi-automatically merged as they are developed. To ensure that language independent and common features and updates do not destroy existing language data or use case, we enforce a rigorous continuous integration based testing regime. The current system for testing is a combination of our long-term investment in testing within the infrastructure locally for developers—combined with modern automatic testing currently supplied by GitHub actions.

The automated testing and integration is one of the key features for upkeep and maintenance of the linguistic data: the linguists work with the dictionaries and rules on a daily basis and receive immediate feedback from the system of the effects of the new word entries or rules. The testing system verifies that if the new words and rules did not affect negatively the user experience of e.g. spelling checker, it can be immediately deployed to the end users of the mobile keyboards and spell-checkers on office platforms.

Another part of the *GiellaLT* philosophy is that of reusable and multi-purposeful resources, cf. Antonsen et al. (2010). This is true for all of our work, from corpus collection to cross-lingual co-operation.

## 2.1 Tools

One of the main aims of the infrastructure is to provide tools to different end user groups: language communities, learners, language users and researchers. In 2012, spell-checking and correction was presented as one of the key technologies that language technology infrastructures can provide as a support tool for linguistic communities. This continues to be a core tool but even it has changed significantly: in 2012, the main use of spelling checkers was most commonly seen as a writer's tool within office suites. While this still is the case, the users will much more likely face spelling correctors as part of e.g. mobile keyboards, in form of automatic corrections. The GiellaLT infrastructure today offer keyboards for many of the languages in the infra for most mobile and computer operating systems. For writer's tools, we also provide more advanced grammatical error correction for some of the languages. This is a tool that in practice concerns sentence level data while correcting errors, whereas spelling checker typically processes at word level mainly. Intelligent dictionaries and corpus resources are provided to users primarily via web apps and related mobile apps. The intelligent dictionaries are an important tool for language learners and users, they enable users to understand texts by looking up the underlying lemma of inflected forms. For research uses as well as for language learners and users to some extent, we also have annotated corpora that can be used for example through a *Korp* corpus webapp. (Borin et al., 2012) Spoken language technology is one of the newer applications in our infrastructure. This encompasses text-to-speech as well as automatic speech recognition.

An overview of the tools available for the languages listed later in the article is given in table 1.

## 2.2 Methods

The foundation for all linguistic processing in the *GiellaLT* infrastructure is the morphological analyser, built using formalisms from Xerox: `lexc`, `xfst` and optionally `twolc`. From these source files, the infrastructure creates *finite state transducers* (FST's) using one of three

| Language | KBD | SP | GC | MT | Dict |
|---|---|---|---|---|---|
| Eastern Mari | B | B | — | — | B |
| Erzya | V | B | — | — | B |
| Faroese | — | V | B | B | — |
| Finnish | — | B | — | B | — |
| Greenlandic | — | V | — | — | V |
| Inari Sámi | V | V | B | B | V |
| Ingrian | B | B | — | — | — |
| Komi-Zyrian | B | B | — | — | B |
| Kven | B | B | — | — | V |
| Livvi | B | B | — | — | V |
| Lule Sámi | V | V | B | B | V |
| Moksha | V | B | — | — | V |
| North Sámi | V | V | V | V | V |
| Norw. bokmål | — | — | — | — | V |
| Norw. Nynorsk | B | B | — | — | — |
| Pite Sámi | — | B | — | — | V |
| Skolt Sámi | V | B | — | — | V |
| South Sámi | V | V | B | B | V |
| Udmurt | B | B | — | — | V |
| Voru | B | B | — | — | V |
| Western Mari | B | B | — | — | V |

Table 1: Tools available for some of the languages in the GiellaLT infrastructure. KBD = Keyboards, SP = spellers, CG = Grammar checker, MT = machine translation, Dict = electronic dictionaries. V = released, B = prerelease.

supported FST compilers: Xerox tools (Beesley and Karttunen, 2003), *HFST* (Lindén et al., 2013), or Foma (Hulden, 2009). All higher-order linguistic processing is done using the VISLCG3 (*visl.sdu.dk*) implementation (Didriksen, 2010) of Constraint Grammar (Karlsson, 1990). Tokenisation is based on an FST model initially presented by Karttunen (2011) in the Xerox tool `pmatch`. The resulting FST is applied using `hfst-tokenise`. In our tokenisation, sentence boundary detection is treated as a special case of ambiguous tokenisation, and solved in the same way, approaching near-perfect sentence boundary identification, cf. Wiechetek et al. (2019b).

Spell-checkers are based on weighted finite-state technology as described by (Pirinen and Lindén, 2014). There is also support for neural network based models of spell-checking (Kaalep et al., 2022), this is however in its current stage still not up to par with the traditional weighted finite-state models given the current error corpus sizes. Since 2019 the *GiellaLT* infrastructure supports building grammar checkers (Wiechetek et al., 2019a) and these are available for some of the Sámi languages already. Another high-level tool available within the *GiellaLT* infrastructure is machine translation. It works in cooperation with the *Apertium* infrastructure (Khanna et al., 2021).

Speech technology is based on a combination of the knowledge-based methods and data-driven methods. For this reason we have started developing workflows and best practices for gathering good spoken data for minoritised and less resourced language scenarios we work with.

The engineering solutions we use to tie together the linguistic work and the technological work follow the contemporary approaches to *continuous integration and deployment*, which at the moment is implemented on *GitHub* systems including GitHub Actions as well as on some custom-built continuous integration systems based on *Tascluster*. The continuous integration tools are used both in the traditional way as in software engineering, to ensure that the new additions to code and data did not fundamentally break the system (e.g. with syntax errors) as well as ensuring the quality of the systems after the change. The quality assurance aspect is based on automated testing of evaluation factors that are both relevant for the products as well as interesting for research and development, e.g. for spell-checkers we test and track the development of *precision and recall* of the system over time.

## 3 Linguistic data

There are two types of linguistic data we gather and develop in the infrastructure, one is the dictionaries, grammars and descriptions for each language and the other is corpus data. Even if our system is not corpus-driven in the way most other contemporary systems are, once we develop the knowledge-based systems we are working for, the real-world data from language users becomes a very important resource for testing and evaluating the systems we have built. The corpus data we collect is also enriched by language experts by annotating spelling and grammar errors with corrections included, or by doing other linguistic annotations and corrections to automated annotations. For this reason and also because we work with many languages that have very little data available the corpora we collect are carefully selected and curated.

The linguistic data can be roughly evaluated without annotated large manually annotated gold corpora by calculating the number of words in the dictionaries and a *naïve coverage*. Words counted are lemma entries, thus words covered by productive morphology will not be included in the

figure.[1] The naïve coverage will give an intuition for the extents of the derivational morphology has with regards to real world word-form usage. Here naïve coverage is calculated as a proportion of tokens that get any analyses of the whole corpus, in this case we use the tokenisation provided by the corpus analysis tools, which is based on left-to-right longest match tokenisation that falls back on space-separated tokens with special cases for punctuation, i.e. mostly natural tokenisation for the western languages with latin and cyrillic scripts. [2] The figures are given in table 2.

| Language | ISO | Words | Coverage |
|---|---|---|---|
| Eastern Mari | mhr | 55 k | 87 % |
| Erzya | myv | 102 k | — |
| Faroese | fao | 72 k | 94 % |
| Finnish | fin | 412 k | 95 % |
| Greenlandic | kal | 12 k | 59 % |
| Inari Sámi† | smn | 77 k | 91 % |
| Ingrian | izh | 2 k | — |
| Komi-Zyrian | kpv | 195 k | 99 % |
| Kven | fkv | 16 k | 75 % |
| Livvi | olo | 58 k | — |
| Lule Sámi† | smj | 76 k | 93 % |
| Moksha | mdf | 41 k | — |
| North Sámi† | sme | 164 k | 91 % |
| Norw. Bokmål | nob | 54 k | 95 % |
| Pite Sámi† | sje | 5 k | 100 % |
| Skolt Sámi† | sms | 66 k | 82 % |
| South Sámi† | sma | 86 k | 84 % |
| Udmurt | udm | 47 k | — |
| Voru | vro | 20 k | 90 % |
| Western Mari | mrj | 26 k | — |

Table 2: Dictionary sizes and coverage for a number of languages in the *GiellaLT* infrastructure; ISO codes are ISO 639-3.
† The figures for some of the Sámi language word counts include 33.5 k proper names in a shared file.

It is noteworthy that the naïve coverages we count are based on the corpora we have collected and this corpora has been seen by people working on the dictionaries, in other words it is technically not a clean test setup. For many of the languages we work with this is necessitated by the facts that the corpus we have is all texts that are available for the language at all. Not making full use of it would hinder the development of the language model in a way that would be more valuable for the language

---

[1]Natural language productive morphology in complex morphologies we work with is usually cyclical, so theoretic word count for derived and compounded forms of all languages is infinite.

[2]c.f. `https://github.com/giellalt/giella-core/blob/master/scripts/coverage-etc.bash`

communities than to hide parts of the corpus from the lexicographers for testing purposes. For this reason the figures should be considered as a rough guideline, as naïve coverage would be anyways. For our intents and purposes, we can see from the naïve coverage if the dictionaries need attention e.g., for spell-checkers to be usable enough as to not show too many red underlines in regular everyday texts.

We collect texts for the Nordic languages as well as several other languages that we use and develop. The largest corpora we have harvested are for the Sámi languages: North, Lule, South, Inari and Skolt Sámi. The Sámi corpus is owned by the Norwegian Sámi parliament, and all corpora are administered and made accessible to the public by the Divvun and Giellatekno groups. The corpora for some of the Uralic languages in Russia are large, and for Meadow Mari even larger than for North Sámi. Some of the corpora for larger, non-minority languages (e.g. Finnish, Norwegian) are moderately sized, since they are already covered by other projects such as OPUS (Tiedemann, 2012), and we only need to create specific corpora for our applications, such as grammar error corpora by L2 speakers in order to develop a grammar checker.

The corpora are split in two based on restrictions set by the copyright owners. Researchers and anyone else can freely download the free part. The whole corpus, also the restricted part, is accessible via a public search interface[3]. We have written a tool named CorpusTools to administer, convert and analyse the corpus texts. Original texts and their metadata are saved in GitHub repositories, then converted to a common XML format, to ease further use of the texts. The sizes of corpora are summarised in table 3, the token count is based on simple space-separated tokens with no extra tokenisation.[4] The languages shown in the table are the Nordic and related languages, for a full listing refer to our website[5]. The corpus texts have some metadata and markups relevant for our use cases, such as grammar checking and correction.

Recently, we have also began collecting speech corpora for speech technology related projects.

For example, for an ongoing Lule Sámi TTS project we reused a part of a Lule Sámi gold corpus from 2013, and collected additional texts we knew to be well written and already proofread, before proofreading these texts once more to avoid confusion when reading the text aloud during the TTS recordings. The Lule Sámi TTS text corpus consists of various text styles (news, educational, parliament etc.) with altogether over 74,000 words. Currently, we have recorded two Lule Sámi voice talents using this text corpus, and after processing the recordings, a speech corpus with altogether 20 hours will be ready to use for speech technology purposes.

| Language | ISO | Tokens | Speech |
|---|---|---|---|
| Eastern Mari | mhr | 57 M | — |
| Erzya | myv | 14 M | — |
| Faroese | fao | 11 M | — |
| Finnish | fin | 2 M | — |
| Greenlandic | kal | 0.5 M | — |
| Inari Sámi | smn | 3 M | — |
| Ingrian | izh | — | — |
| Komi-Zyrian | kpv | 1 M | — |
| Kven | fkv | 0.5 M | — |
| Livvi | olo | 0.3 M | — |
| Lule Sámi | smj | 2 M | 20 h |
| Moksha | mdf | 13 M | — |
| North Sámi | sme | 39 M | 38 h |
| Norw. bokmål | nob | 14 M | — |
| Norw. Nynorsk | nno | 0.7 M | — |
| Pite Sámi | sje | — | — |
| Skolt Sámi | sms | 0.25 M | — |
| South Sámi | sma | 2 M | — |
| Udmurt | udm | — | — |
| Voru | vro | 0.67 M | — |
| Western Mari | mrj | 6 M | — |

Table 3: Corpus sizes for some of the languages in our infrastructure. Tokens are space-separated tokens.

As spoken language technology is based on data and machine learning, the procedures and pipelines described above could be applied to any (minority) language with a low-resource setting, in the task of developing speech technology applications. Most of the applications discussed here can be piloted with or further developed with relatively small data sets (even with < 5 hrs of paired data), compared to the amounts of data used for respective tools for majority languages (see, e .g., Ito and Johnson (2017)[6]). This is largely possible thanks to the available open source materials and technologies, especially those relying on, e.g., *transfer*

---

[3]gtweb.uit.no/korp (Sámi), gtweb.uit.no/f_korp (Baltic Finnic and Faroese), gtweb.uit.no/u_korp (other Uralic languages). Cf. also More info about the corpora.

[4]The corpora are being constantly harvested, the status as of 2023-02-03 is shown, the current status will be available in our GitHub repositories in the near future.

[5]https://giellalt.github.io/

---

[6]The LJ Speech dataset consists of 13,100 short audio clips of a single English speaker with a total length of approximately 24 hours.

*learning*, i. e. fine-tuning of models (Fang et al., 2019).

## 4 Conclusion

In this article we have presented recent developments and status of the *GiellaLT* Nordic multilingual infrastructure built during the last 20 years. In the last years, we have added more support to speech technologies, and keyboards for various platforms such as mobile devices and modern operating systems.

The *GiellaLT* infrastructure contains building blocks and support for most of the language technology needs of indigenous and minority languages, from the very basic input technologies like keyboards to high-level advanced tools like world-class grammar checking and machine translation. It does this by using rule-based technologies that makes it possible for any language community to get the language technology tools they want and need. All that is needed is a linguist.

We discussed the ways for long-term maintenance of linguistic data and software tools for NLP of Nordic and minority languages. We showed some best current practices and workflows on how to maintain the lexicons and keep end user tools unbroken and still up-to-date.

In conclusion, building corpora is based on big efforts, requires expertise and is time-costly. We have illuminated the work behind three important steps within building corpora - firstly, collecting and digitalising, secondly upgrading, i.e. adding annotation for special purposes, and proofreading, and thirdly converting from one medium/language to another as in recording speech, translating, or other.

With our multilingual infrastructure and our language resources we show that while there is a need for corpus data for certain tasks, high quality tools needed by a language community can be built time-efficiently without big data in a rule-based manner.

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
