# OpenReview forum: "GiellaLT---a stable infrastructure for Nordic minority languages and beyond"
_NoDaLiDa/2023/Conference — NoDaLiDa 2023_

### Official Review · Reviewer_HQuz · 2023-03-10
**Good overview of GiellaLT infrastructure**

**Rating:** 7
**Confidence:** 4

**Review:**

The paper describes the Nordic language technology infrastructure, which has been developed for 20 years. Authors give an overview of language resources involved in the project. Paper explains how important it is to keep the linguistic and technical issues separate especially for unresourced, minoritised languages.
Unfortunately, the article remains general, and it would have been helpful if the paper would have a table describing the language technology tools are used for particular languages (for example, is there a spellchecker, grammar checker, tagger, parser, etc. for the given language). The paper is an overview paper and a good fit for a poster or demonstration paper.


**Paper Type:**

Short paper

---

### Official Review · Reviewer_N8QK · 2023-03-10
**Presentation of the adds-ons to the GiellaLT platform**

**Rating:** 6
**Confidence:** 3

**Review:**

This paper presents updates on an existing Nordic multimedia platform GiellaLT. The authors describe how they have added more support to speech to text and text to speech, and keyboards for different devices and other operating systems. While I believe that the platform is important for Nordic indigenous languages as Sámi and other under resourced languages, I cannot see a clear scientific contribution in this paper. It feels more of a report rather than a scientific contribution. I have no doubt that such a platform will advance state-of-the-art approaches and resources for e.g. the Sámi language, but I lack a more technical description of the platform.

**Paper Type:**

Short paper

---

### Decision · Program_Chairs · 2023-03-17

Accept